# Direct-to-biology, automated, nano-scale synthesis, and phenotypic screening-enabled E3 ligase modulator discovery

Zefeng Wang[1,8], Shabnam Shaabani [1,8], Xiang Gao[2,8], Yuen Lam Dora Ng [3], Valeriia Sapozhnikova[3,4,5], Philipp Mertins [5,6], Jan Krönke [3,4] ✉ & Alexander Dömling [1,7] ✉

Thalidomide and its analogs are molecular glues (MGs) that lead to targeted ubiquitination and degradation of key cancer proteins via the cereblon (CRBN) E3 ligase. Here, we develop a direct-to-biology (D2B) approach for accelerated discovery of MGs. In this platform, automated, high throughput, and nano scale synthesis of hundreds of pomalidomide-based MGs was combined with rapid phenotypic screening, enabling an unprecedented fast identification of potent CRBN-acting MGs. The small molecules were further validated by degradation profiling and anti-cancer activity. This revealed E14 as a potent MG degrader targeting IKZF1/3, GSPT1 and 2 with profound effects on a panel of cancer cells. In a more generalized view, integration of automated, nanoscale synthesis with phenotypic assays has the potential to accelerate MGs discovery.

Targeted protein degradation (TPD) is an emerging therapeutic modality and has attracted great attention from academia and industry[1,2]. The prototypical TPD agents, molecular glues (MGs) and proteolysis targeting chimeras (PROTACs), can lead to temporal proteasomal degradation of the protein-of-interest (POI). PROTACs are small heterobifunctional molecules integrating an E3-ligase binder and a POI binding moiety through a synthetic linker construct. The PROTACs technology has been applied to degrade numerous pathological proteins and a rich pipeline is currently progressing into preclinical and early clinical trials[3-5]. However, overcoming PK/PD issues towards clinical compounds is demanding due to the intrinsically high molecular weight and related physicochemical properties[6]. On the other hand, MGs are small molecules with beneficial 'drug-like' physicochemical properties binding to an E3 ligase, and, similarly to PROTACs, leading to neosubstrate proteasomal degradation. Their mechanism of action is however less predictable; their often hydrophobic surface-exposed portions of the MGs seem to change the hydrophobic surface area of the E3 ligase and thereby leading to neosubstrate ubiquitination and degradation[7,8]. MGs have already proven their validity as marketed drugs, as there are several approved drugs or clinical compounds working by an MG mechanism (Fig. 1A), for example, the IKZF1/3 degrader thalidomide and its analogs pomalidomide and lenalidomide[8], and the RBM39 degrader indisulam[9]. Thalidomide analogs induce selective ubiquitination and degradation of two lymphoid transcription factors, IKZF1 and IKZF3, by the CRBN-CRL4 ubiquitin ligase[10]. Additionally, CSNK1A1 (CK1α) was recently discovered as a lenalidomide-specific neo-substrate[11]. Interestingly, modification of pomalidomide or lenalidomide can have a profound impact on the degradation potency and degradation profiles. For example, CC-220 (Fig. 1A) showed 10-fold more potency in the cells than lenalidomide,

[1]University of Groningen, Department of Drug Design, A. Deusinglaan 1, 9713 AV Groningen, The Netherlands. [2]Department of Internal Medicine III, University Hospital Ulm, 89081 Ulm, Germany. [3]Department of Hematology, Oncology and Cancer Immunology, Charité - Universitätsmedizin Berlin, corporate member of Freie Universität Berlin and Humboldt-Universität zu Berlin, Berlin, Germany. [4]German Cancer Consortium (DKTK) partner site Berlin and German Cancer Research Center (DKFZ), Heidelberg, Germany. [5]Max Delbrück Center for Molecular Medicine, Berlin, Germany. [6]Berlin Institute of Health, Berlin, Germany. [7]Institute of Molecular and Translational Medicine, Faculty of Medicine and Dentistry and Czech Advanced Technology and Research Institute, Palackȳ University in Olomouc, Olomouc, Czech Republic. [8]These authors contributed equally: Zefeng Wang, Shabnam Shaabani, and Xiang Gao. ✉e-mail: jan.kroenke@charite.de; alexander.domling@upol.cz

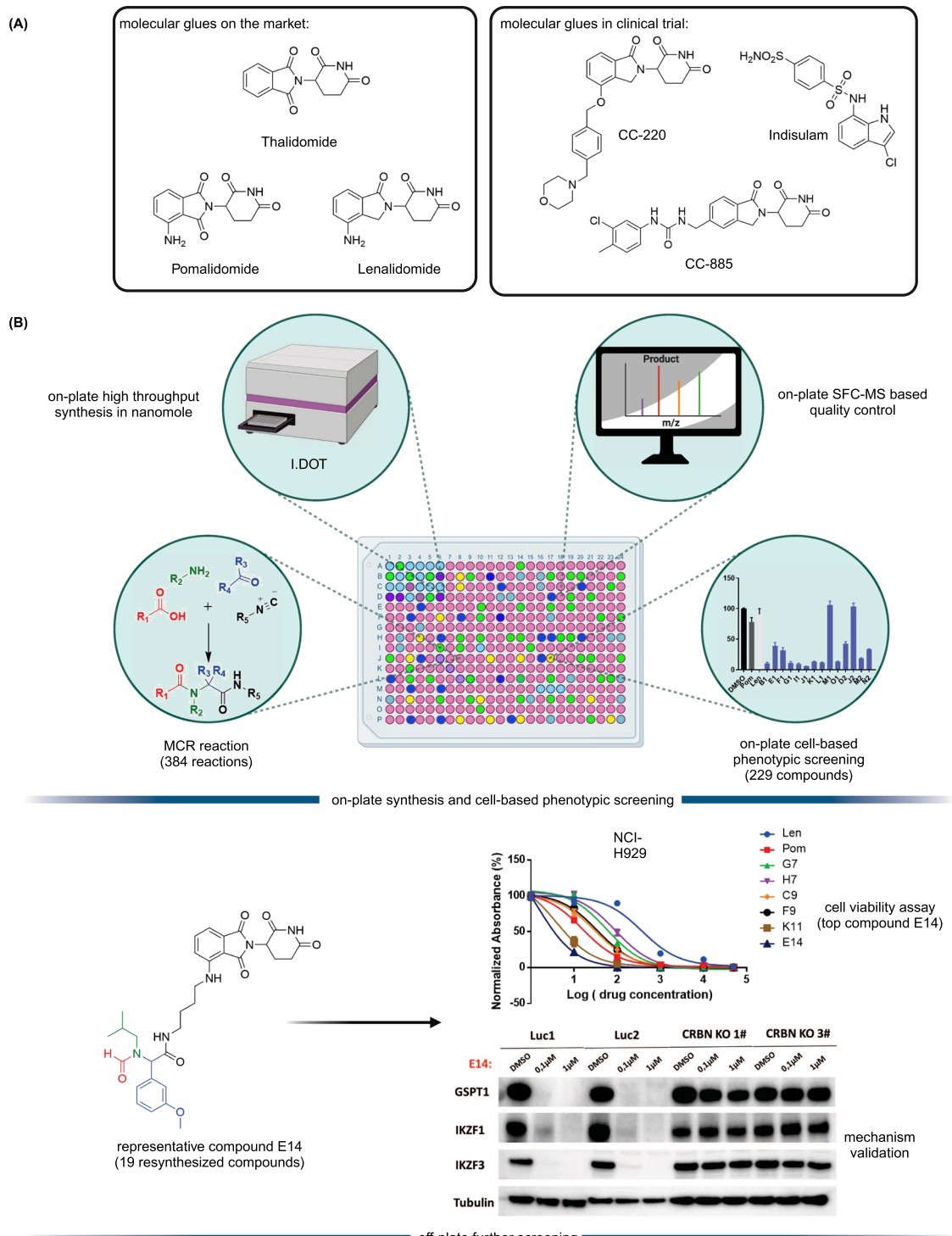

**Fig. 1 | Representative reported MGs and D2B workflow. A** chemical structures of representative MGs. **B** Direct-to-biology platform combines high-throughput nano scale synthesis and phenotypic screening to accelerate E3 ligase modulator discovery. I.DOT means Immediate Drop on Demand Technology. Created with BioRender.com.

and CC-885 (Fig. 1A) was found to induce degradation of the substrate GSPT1[12,13]. Both MGs and PROTACs are emerging drug modalities providing interesting features over classical pharmacology-driven drugs by their ability to drive the destruction of proteins that have multiple functions, thereby potentially overcoming resistance mechanisms and providing new pharmacology. While PROTACs can be developed highly rationally, MGs are discovered rather serendipitously requiring synthesis and testing of large series of compounds[14,15]. Additionally, the discovery of MGs and PROTACs is done in a sequential, often mmol scale synthesis which is time-consuming and expensive.

In this work, to address current shortcomings in MGs discovery, we use the direct-to-biology (D2B) approach and combined the automated, high throughput miniaturized synthesis with cell-based phenotypic screening (Fig. 1B). The I.DOT (Immediate Drop on Demand Technology, a pressure-based nano dispensing technology) is employed to accelerate the synthesis of diverse MGs libraries on nano scale[16–21]. In a subsequent cell-based phenotypic screening cascade, the

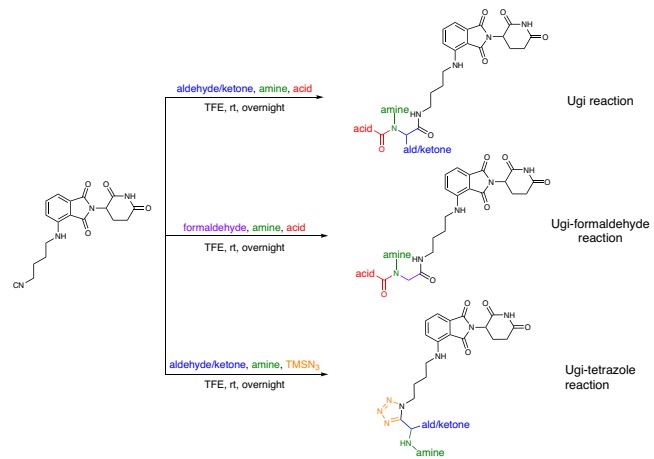

**Fig. 2 | Isocyanide-based multicomponent reactions.** Ugi reaction, Ugi-formaldehyde reaction and Ugi-tetrazole reaction.

compounds are tested in the thalidomide and analog sensitive MM.1S multiple myeloma cell line which reportedly is used for MGs screening[22]. In this D2B screening platform, the crude compounds are directly screened on cells without further chromatographic purification or clean up. Then, the 19 best compounds are selected for re-synthesis on mmol scale followed by purification and fully characterized. Next, the purified compounds are rescreened in a panel of cell lines with differential CRBN status, further supporting their mechanism of action. Protein analysis reveals the degradation targets and potency of the different compounds and identifies E14 as a new potent degrader of IKZF1, IKZF3, GSPT1, and GSPT2.

## Results

### Nano-scale synthesis and destination plate analysis

To build a highly diverse library around the pomalidomide pharmacophore, we chose two different isocyanide-based multicomponent reactions (IMCRs). IMCRs are well known to yield diverse drug-like scaffolds and have been employed for the synthesis of multiple biologically active molecules[23]. Advantageously, as opposed to classical multi-step linear syntheses, IMCRs provide a complex product molecule in just one step. In this project, the Ugi 4-component reaction (U-4CR) of oxo components, primary amines, acids, and isocyanides, the Ugi-formaldehyde reaction (formaldehyde is used as the oxo component in the U-4CR), and the Ugi-tetrazole reaction (UT-4CR) of oxo components, amine, TMSN3, and isocyanide were used to synthesize the MGs library (Fig. 2). The isocyanide in IMCRs is often the limiting component, due to its reduced availability. Thus, we prepared a pomalidomide-derived isocyanide (Supplementary Table. 1). Stock solutions of the aldehyde/ketone (including formaldehyde 37% w/w aq solution), amine and carboxylic acid building blocks were prepared as 0.5 M trifluoroethanol (TFE), the isocyanide as 0.25 M TFE (due to its limited solubility) and the TMSN3 as 0.6 M TFE (Fig. 3B). The building blocks were chosen according to diversity criteria: the oxo building blocks (aldehydes and ketones) consisted of aromatic, heteroaromatic, alicyclic and aliphatic linear (branched) motives. Additionally, halogen, trifluoromethyl, hydroxyl, methyl ether, and tertiary amine functional groups were incorporated. Similarly, the diversity of the other components, carboxylic acids and primary amines was kept broad. All the building blocks were pipetted into a 96-well source plate shortly before the reaction (Fig. 3A). For U-4CR (wells A1-F8 on destination plate), amine building blocks (1 eq, 125 nL), aldehyde/ketone building blocks (1 eq, 125 nL), carboxylic acids (1 eq, 125 nL) and isocyanide (1 eq, 250 nL) were transferred to the destination plate (384 wells format) by I.DOT, respectively. In the wells F9-K16 of destination plate, formaldehyde aq solution was used as oxo component in U-4CR.

For UT-4CR (wells K17-P24 on destination plate), amine building blocks (1 eq, 125 nL), aldehyde/ketone building blocks (1 eq, 125 nL), isocyanide (1 eq, 250 nL) and TMSN3 (2.4 eq, 250 nL) were transferred, respectively. Using I.DOT technology, it takes <20 mins to transfer all four components to the 384-well destination plate. A randomizing algorithm allowing for sparse matrix synthesis previously developed by us, was used in order to produce a highly diverse library of products. Theoretically, the chemical space corresponds to 23,520 products based on the three IMCRs variations using 87 different starting materials. In our project, a small subset of 384 compounds were produced for the test. When the starting material transfer was done, the destination plate was sealed and kept at room temperature for 24 h on an orbital shaker. Three product scaffolds (Fig. 3B) could be generated in a high throughput fashion using I.DOT and all the product structures are shown in Supplementary Table. 2.

Next, the reaction quality was analyzed by mass spectrometry (MS) as previously described by us[18,24,25]. Shortly, 30 μL ethylene glycol was added into each well and the sample was analyzed by direct injection into a mass spectrometer. Using our in-house program to analyze the mass data, each well was designated as green (major product formation) if the peak corresponding to [M + H], [M+Na], or [M + K] was the highest peak, if the corresponding peak did not exist, the well was designated as blue (no product formation); and otherwise as yellow (medium product formation)[18]. The heat map describing the overall reaction success is depicted in Fig. 4A (Supplementary Fig. 1 and Fig. 3). Overall, 60% of the reactions showed the product peak on the mass spectrometer (green and yellow wells), while 40% reaction showed no product formation (blue wells). The Ugi-formaldehyde reaction performed the best with 67% of the wells showing product formation (Supplementary Fig. 2). The Ugi-tetrazole and Ugi reaction also worked well with 61% and 51% showing the product formation, respectively. We also analyzed the performance of the different building blocks in different reactions (Supplementary Figs. 3–9). The different and diverse building blocks were employed in the IMCRs, which makes the rapid survey of the reactivity of the building blocks and several scaffolds at once possible. In the Ugi reaction, aldehydes perform better than the ketones, especially aromatic aldehydes, such as S-O9, S-O15, S-O25, S-O27 (S means 96-well source plate). Noteworthy, many compounds with polar functional groups worked very well in Ugi reaction, including morpholine (S-A10), pyridine (S-O27, S-A4), cyano group (S-A9). This is important to keep control over the lipophilicity of the final products. A similar functional group tolerance was also found in the Ugi-formaldehyde reaction where the amine building blocks with polar group performed very well, e.g., pyridine (S-A4, S-A6, S-A7), morpholine (S-A10). For the carboxylic acid building block, almost all of them performed well in the Ugi-formaldehyde reaction, except S-C5 and S-C23 which showed no product formation. Both aromatic and aliphatic carboxylic acids were well tolerated.

### Phenotypic screening of the destination plate

Phenotypic screening is a strategy successfully employed in drug discovery to identify the molecules with the ability to alter the phenotype of the cells[26,27]. Cell-based phenotypic screening assay is one of the most important strategies used to identify active molecules. The wells which showed product formation based on the mass spectrometer analysis (229 wells designated as green and yellow in Fig. 4) were selected for further phenotypic screening. MM.1S is a representative cell line of multiple myeloma lineage that is highly dependent on IKZF1 and IKZF3 and therefore, commonly used in CRBN MGs screening. Most of the compounds were potent in the on-plate MM.1S cell line screening (Supplementary Fig. 10). For instance, 73% (167 wells) of wells indicated strong inhibition (cell viability <50%) and higher potency than the positive controls, pomalidomide and lenalidomide.

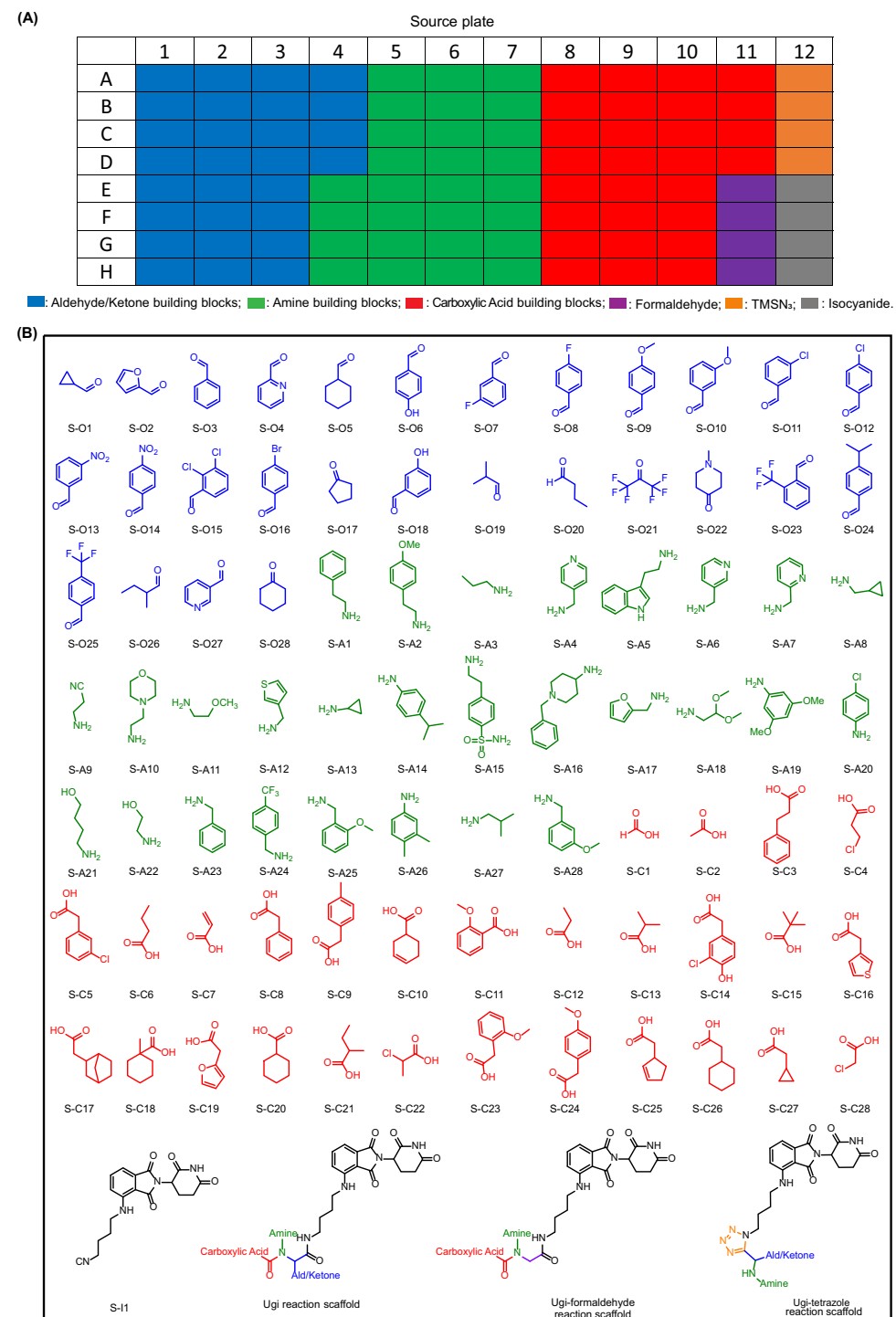

**Fig. 3 | Source plate layout and chemical structures of building blocks. A** Source plate layout. **B** Building block chemical space: aldehydes/ketones, amines, carboxylic acids, isocyanide and three MCR product scaffolds. S means source plate.

## Biological evaluation of resynthesized compounds with different cell lines

Next, we resynthesized and purified 19 compounds (Fig. 4B). They were picked and selected based on their performance in the phenotypic screening (Supplementary Fig. 10). They all showed high inhibition against MM.1S cells. Meanwhile, the compound diversity was also considered. For example, considering the moiety from amine component, not only aromatic amines but also aliphatic amines were selected. The purified compounds were rescreened in the MM.1S cell line. Almost all resynthesized compounds showed high activity

compared to the positive control pomalidomide and lenalidomide at a concentration of 1.0 μM in MM.1S cell line (Fig. 5A). For instance, 12 compounds (F11, G13, H12, H23, J7, K7, L13, L18, M11, N9, O9, P24) showed moderate inhibition. Out of these 12 compounds, five compounds (F11, J7, L13, L18, N9) performed similarly to pomalidomide, with an inhibitory effect of >50% at 1.0 μM. Top six compounds (C9, E14, F9, G7, H7, K11) showed a higher inhibition effect than pomalidomide with E14 being the most potent. These results are in accordance to the initial screen and underline the validity of our phenotypic screening D2B platform. The top 6 compounds were selected for

**(A)**

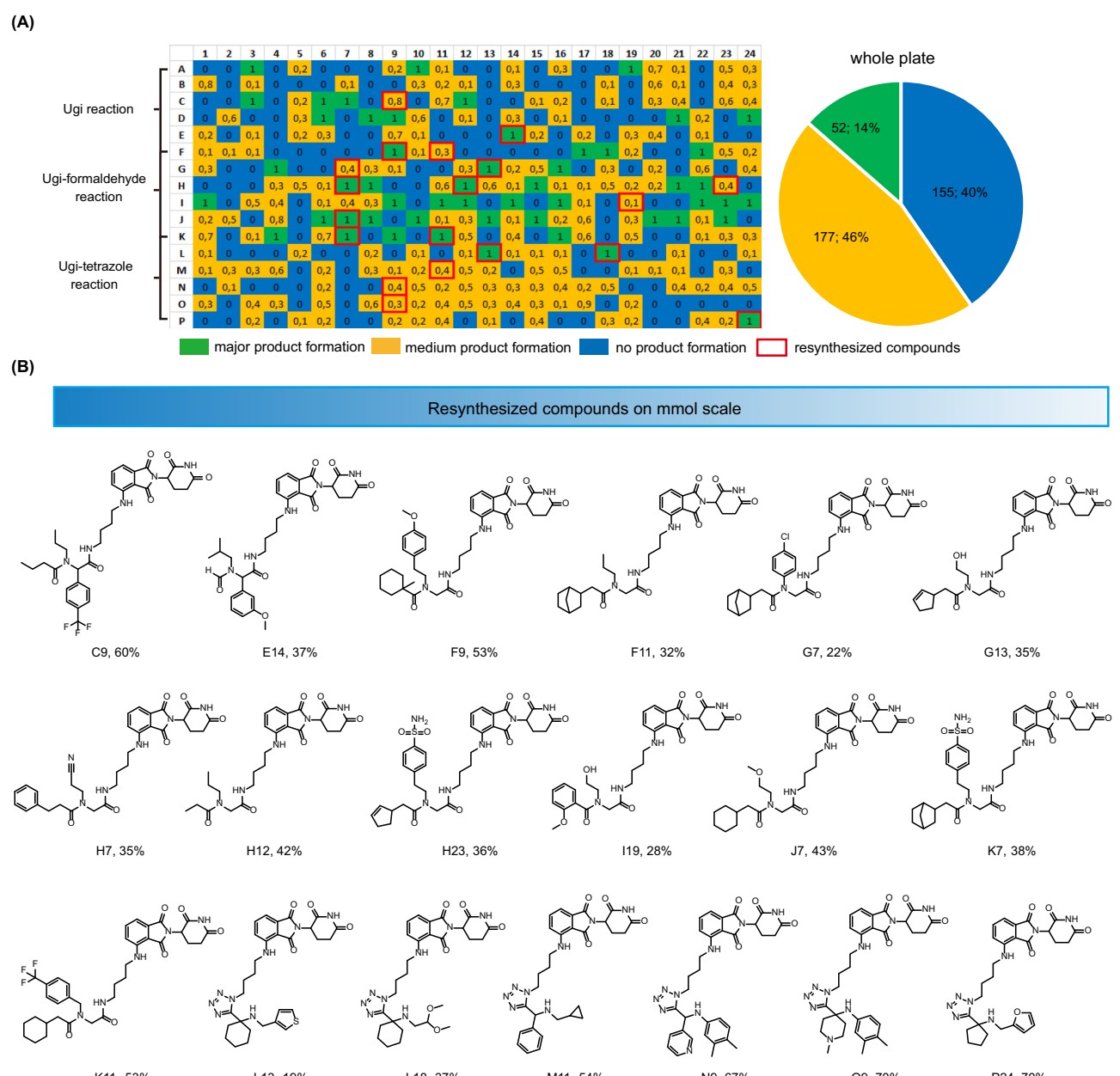

**(B)**

Fig. 4 | Destination plate analysis and resynthesized compounds. A Reaction layout and heat map of the 384-well destination plate based on the mass analysis (with normalized values in each well). Pie chart of the overall synthesis success of the 384-well plate. B Resynthesized compounds with isolated yield. Source data are provided as a Source Data file.

further studies (Fig. 5B). All 6 compounds showed inhibition in a concentration-dependent manner and performed better than lenalidomide (EC$_{50}$ = 705.1 nM (MM.1S) and EC$_{50}$ = 373.0 nM (NCI-H929), respectively). Two compounds were more potent than pomalidomide (EC$_{50}$ = 52.53 nM) in the MM.1S cells (EC$_{50}$ (K11) = 43.79 nM, EC$_{50}$ (E14) = 11.08 nM) and NCI-H929 cells (EC$_{50}$ (K11) = 3.9 nM, EC$_{50}$ (E14) = 1.485 nM, and EC$_{50}$ (pomalidomide) = 15.65 nM, respectively). To determine whether the observed effects are based on IKZF1/3 degradation or other substrates, we screened the compounds with high anti-proliferative activity in the human leukemia cell lines HEL and K562, which are IKZF1/3 independent and lenalidomide- and pomalidomide-insensitive (Fig. 5C)[8]. Compared to the controls, almost all the compounds effectively inhibited HEL cell growth in a concentration-dependent manner and E14 also showed strong inhibition at 1.0 µM in K562 cells, which were unexpected results. Taken together, these experiments indicate that all 6 compounds have anti-proliferative activity in IKZF1-dependent and independent cell lines in a concentration-dependent manner, indicating that other substrates are degraded by these compounds that account for the toxicity in cancer cells.

**Protein degradation profile investigation**

We next investigated the effect on neo-substrates of the top 6 compounds. Since all compounds in the library contained the conserved pomalidomide motif, we anticipated comparable binding affinities to the CRBN − DDB1 complex. MM.1S cells were treated with the 6 compounds at two concentrations (1.0 µM and 10.0 µM) for 18 h. Indeed, all 6 compounds from the library induced IKZF1/3 degradation, with G7, F9, and E14 being more potent than pomalidomide and lenalidomide (Fig. 6). Surprisingly, all 6 compounds also lead to the degradation of

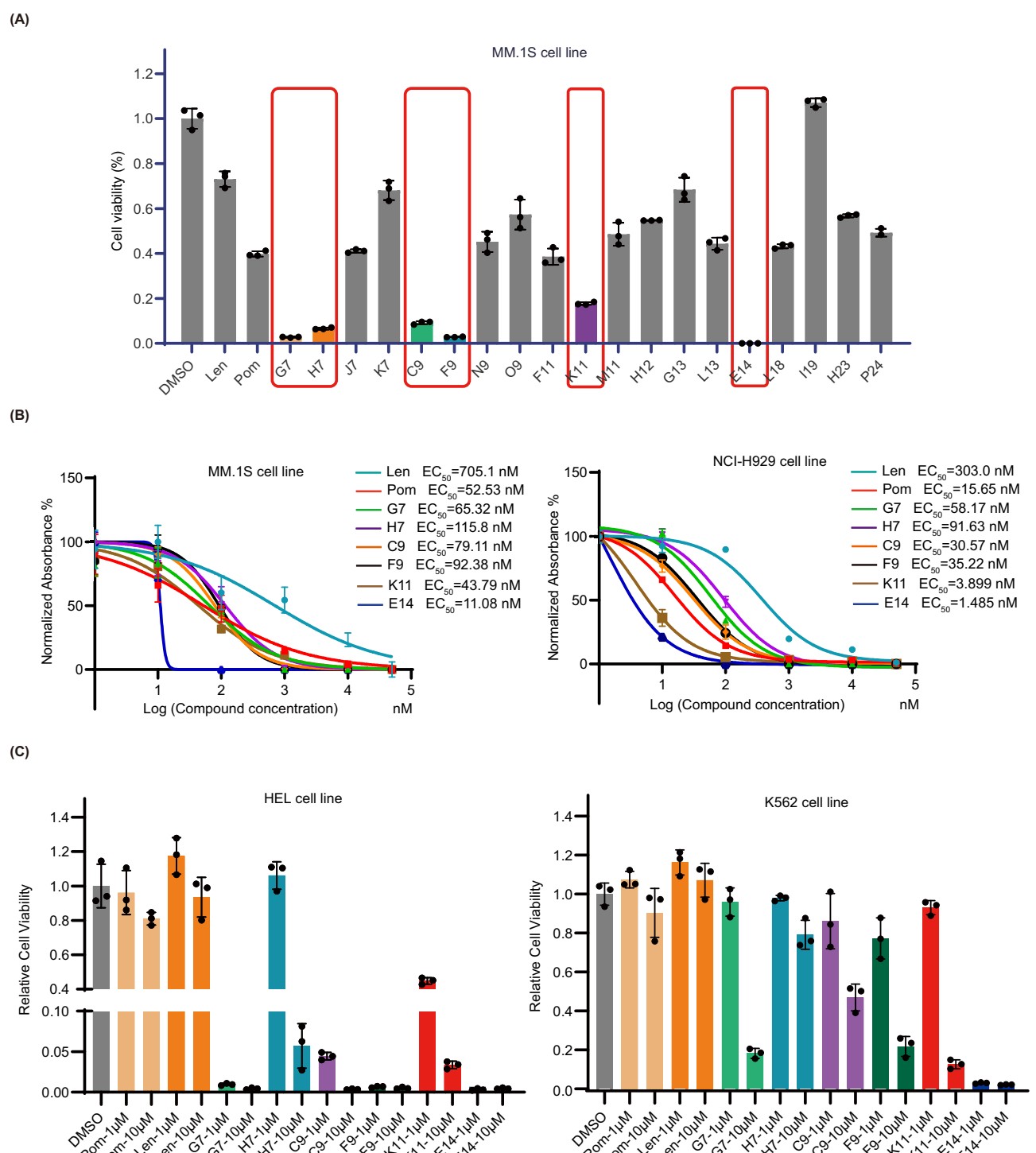

**Fig. 5 | Cell viability assays with selected compounds in various cell lines.**
**A** MM.1 S cell viability assay at 1.0 μM for 96 h. Graph bars represent mean values ± SD. $n = 3$ independent biological replicates. **B** Anti-proliferative activity assay in MM.1S and NCI-H929 cell lines treated for 96 h. Graph bars represent mean values ± SD. $n = 3$ independent biological replicates. **C** cell viability assay at 1.0 μM and 10.0 μM in IKZF1/3 independent cell lines (HEL and K562 cell lines) for 96 h. Graph bars represent mean values ± SD. $n = 3$ independent biological replicates. Source data are provided as a Source Data file.

GSPT1 in contrast to pomalidomide and lenalidomide, which could be a possible explanation of their anti-proliferative effects in IKZF1/3-independent cell lines.

To further validate the degradation mechanism, we constructed CRBN knockout (KO) MM.1S and RPMI-8226 cell lines by using CRISPR/Cas9 to explore whether the toxicity and degradation depends on the presence of CRBN. In cell viability assays, all 6 synthesized and 2 positive control compounds were tested in wild-type and CRBN

knockout MM.1S and RPMI-8226 cell lines. The experiment results from MM.1S (Fig. 7A) apparently showed that in the absence of CRBN, the anti-proliferative activities of the compounds were abolished, further confirming a CRBN-dependent mechanism. Not surprisingly, the RPMI-8226 cell viability assay (Supplementary Fig. 11) indicated the same conclusion as MM.1S cell line. In addition, immunoblot analysis was conducted to verify the degradation difference between wild-type and CRBN KO cell lines. None of the tested compounds could induce

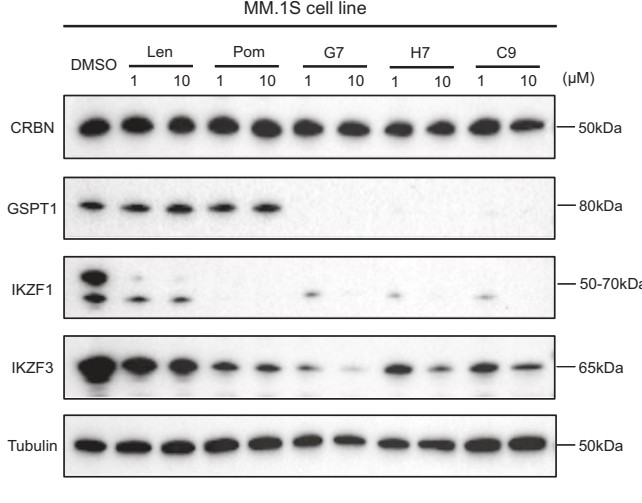

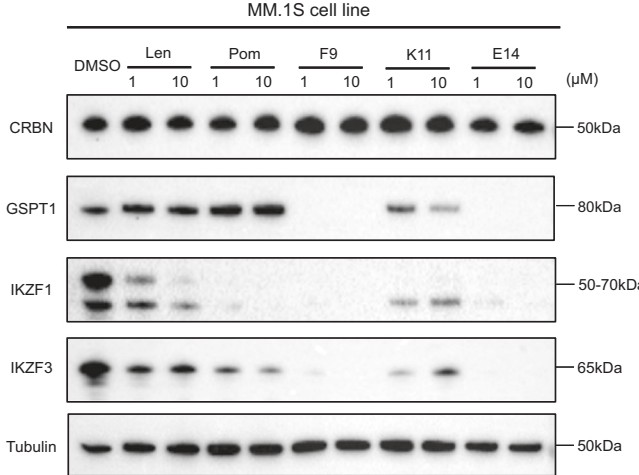

**Fig. 6 | Immunoblots in MM.1S cells after 18 h treatment for all six tested compounds.** Pomalidomide and lenalidomide are shown for comparison. The experiment was repeated twice independently with similar results. Source data are provided as a Source Data file.

neosubstrate degradation in the absence of CRBN (Fig. 7B), confirming that they act through CRBN-mediated degradation.

### Comparison between E14, pomalidomide, and CC-885 in anti-proliferative activity and degradation efficiency

Since our compounds were able to induce GSPT1 degradation, CC-885, an MG known to potently degrade GSPT1, was taken as a reference to compare with our most potent compound (E14) in the cell viability assay. E14 showed similar inhibition with CC-885 against the tested cell lines (Fig. 8A). Notably, E14 and CC-885 both decreased cell viability of IKZF1/3-independent HEL and K562 cells. The degradation of GSPT1 in MM.1 S cells induced by E14 and CC-885 were confirmed via western blotting. In Fig. 8B, after 16 h treatment, E14 induced degradation of GSPT1 with $DC_{50} = 0.88$ nM, IKZF1 with $DC_{50} = 11.54$ nM, and IKZF3 with $DC_{50} = 13.84$ nM.

### The effect of compound E14 on the proteome of MM.1S cells

To investigate the effects of compound E14 on cellular protein levels more broadly and evaluate its overall proteome-wide selectivity, we performed bottom up mass spectrometry-based proteomic analysis in MM.1S cells treated for 18 h at a 1 μM concentration. Compound E14 led

to potent downregulation of IKZF3, GSPT1, and remarkably also GSPT2 which has high homology (97%) with GSPT1 but less frequently reported as IMiD neo-substrate (Fig. 9). Additional known IMiD neo-substrates ZFP91 and IKZF1 showed a lower but still significant decrease[28]. Several other proteins showed deregulation at a lower level in both directions and likely include many downstream effects given the high toxicity of E14 on cells.

## Discussion

MGs and PROTACs are emerging drug modalities mostly investigated as anti-cancer drugs[29]. Although PROTACs and MGs in principle have a highly similar mechanism of action inducing temporal control over a target protein, from a drug discovery standpoint, MGs are advantageous as they possess 'drug-like' properties such as MW, cLogP, and number of HBD and HBA, increasing their chance of oral bioavailability[30]. Thus MGs are preferable over PROTACs as a drug modality. MGs often have been discovered by serendipity, while few rational discovery strategies are being developed[15,31]. Although platforms for rapid synthesis of PROTACs were reported, pipelines for MGs rapid development were rarely described[32–34]. Here, we describe a convenient D2B approach to discover MGs, combining automated miniaturized synthetic chemistry together with a phenotypic screening. Using nanoscale, I.DOT-enabled synthesis in 384 well plates, we directly screened the crude compounds on a IKZF1/3-dependent cell line. The most potent hits were resynthesized, rigorously purified, and further characterized in more time and work intensive protein degradation assays. Thalidomide derivatization is a viable strategy for targeted protein degradation and demonstrates human degradation pharmacodynamics with clinical E3 ligase modulating drugs[35,36]. For the HT chemistry, we synthesized a pomalidomide pharmacophore-based isocyanide which was used as a fixed starting material in three multicomponent reactions to synthesize sparce matrix arrays of diverse compounds. Thus, we discovered the exceedingly potent compound E14. Compared to pomalidomide, E14 shows enhanced proteasome-dependent degradation of IKZF1/3 and GSPT1 as well as its close homolog GSPT2 which has so far only been described as IMiD neo-substrate in two other publications[37,38]. E14 has also broader and enhanced anti-proliferative activity across a large set of cancer cell lines. Direct screening of the crude reaction compounds was recently described in several medicinal chemistry projects as a strategy to accelerate screening of a large chemical space[39,40]. Additionally, we predict that highly miniaturized automated chemistry of a diverse chemical space together with a D2B screening approach, followed by resynthesis of the hits and rescreening has the potential to accelerate MG discovery[41].

## Methods

### Reagents

Pomalidomide (P0018, Sigma-Aldrich company), lenalidomide (SML2283, Sigma-Aldrich company) and CC-885 (HY-101488, MedChemExpress) were purchased. All other tested compounds were synthesized, and the procedures are described in the Supplementary information. All drugs were dissolved in anhydrous dimethylsulfoxide (DMSO, EC number 200-664-3; Sigma-Aldrich company) at 10 mM, stored at −20 °C as stock solutions, and diluted to the respective concentrations for in vivo experiments.

### Cell culture

MM.1S, NCI-H929, RPMI-8226, HEL, K562, Nalm-6 cells were obtained from the American Type Culture Collection (ATCC) or the Deutsche Sammlung von Mikroorganismen und Zellkulturen (DSMZ). Cells were cultured in RPMI1640 medium (Gibco) or DMEM (Gibco) supplemented with 10% fetal bovine serum (FBS) (Merck Millipore), 1% Penicillin/streptomycin (Gibco), and 1% L-Glutamine (Pan Biotech).

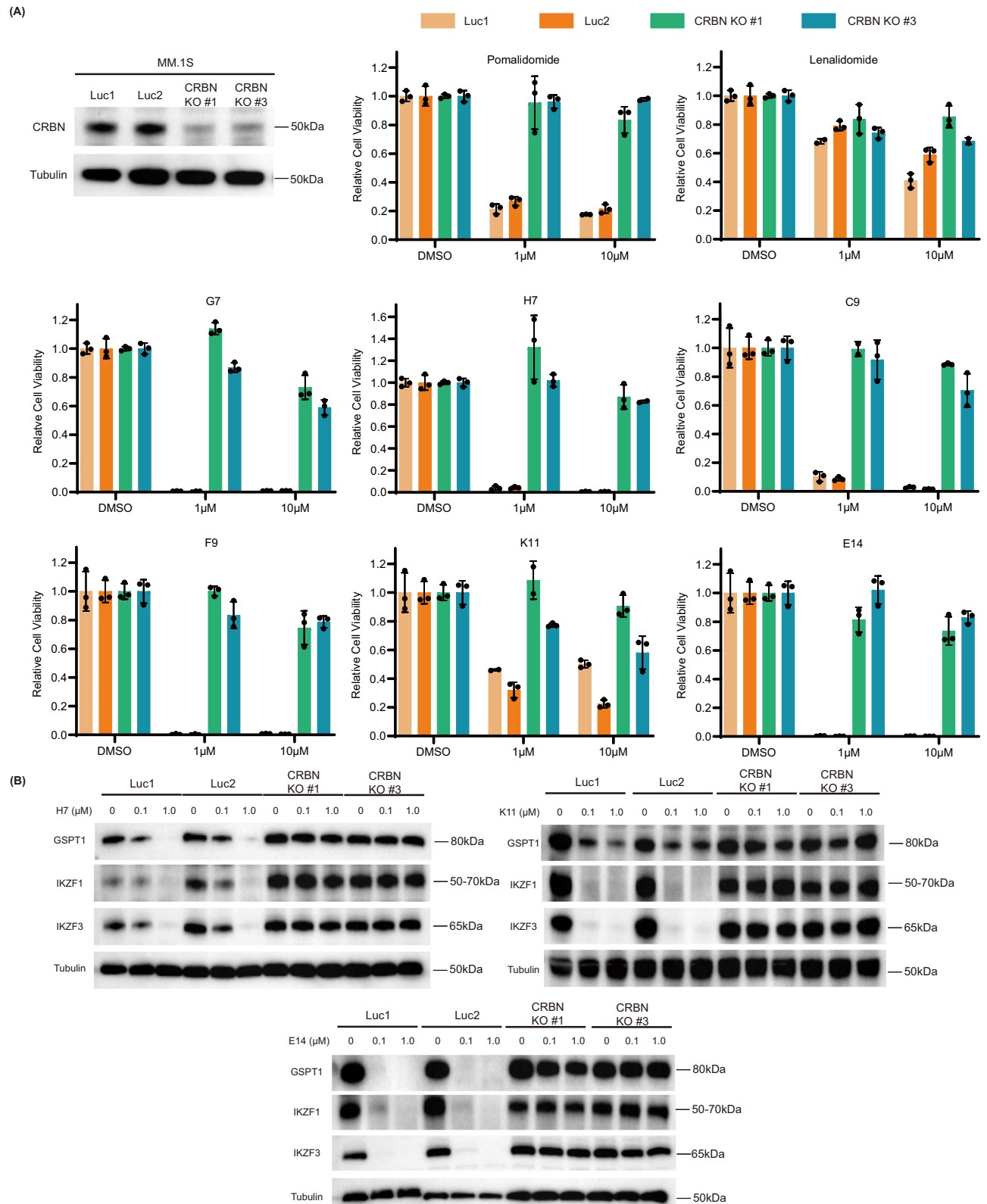

**Fig. 7 | CRBN inactivation abrogates effects of the MGs on toxicity and neo-substrate degradation. A** Cell viability assay in MM.1S and CRBN KO MM.1S cell lines for 96 h at different concentrations. sgRNAs targeting luciferase (Luc1 and Luc2) are used as control. Graph bars represent mean values ± SD. *n* = 3 independent biological replicates. **B** Immunoblot analysis for neo-substrates in the MM.1S and CRBN KO MM.1S cell lines treated with H7, K11, and E14 for 18 h. sgRNAs targeting luciferase (Luc1 and Luc2) are used as control. The experiment was repeated three times independently with similar results. Source data are provided as a Source Data file.

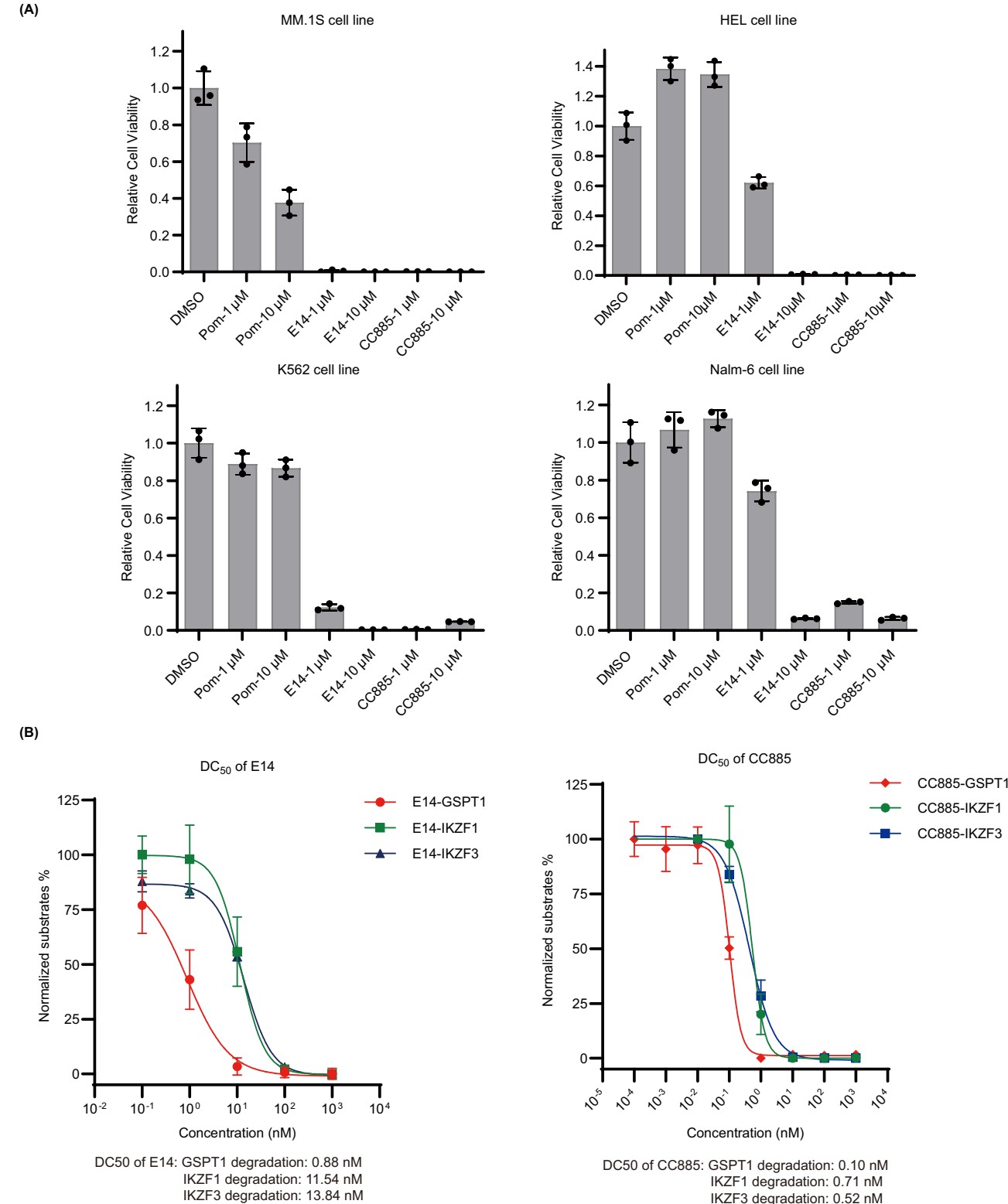

**Fig. 8 | Anti-proliferative activity and degradation efficiency comparison. A** The comparison of cell viability assay between K11, E14, pomalidomide, and CC-885 in different cell lines. Graph bars represent mean values ± SD. *n* = 3 independent biological replicates. **B** Degradation of neo-substrates by E14 and CC-885 in MM.1 S cells after 16 h treatment. Graph bars represent mean values ± SD. *n* = 3 independent biological replicates. Immunoblot experiment was performed to check protein level. Quantitative protein analysis was performed by software Image J. Source data are provided as a Source Data file.

NCI-H929 cells were cultured in the presence of 0,05 mM 2-Mercaptoethanol (Milipore). Cells were grown at 37 °C with 5% $CO_2$ in humidified atmosphere. All cells were cultured in proper density and split every 2–3 days.

**Genetic inactivation by CRISPR/Cas9 in cell lines**

MM.1S, RPMI-8226 cell lines stably expressing Cas9 were generated by using pLKO5d.SSFV.SpCas9.P2a.BSD and selected by Blasticidin (10 μg/ml) (ant-bl, InvivoGen). Single-guide RNAs targeting CRBN were

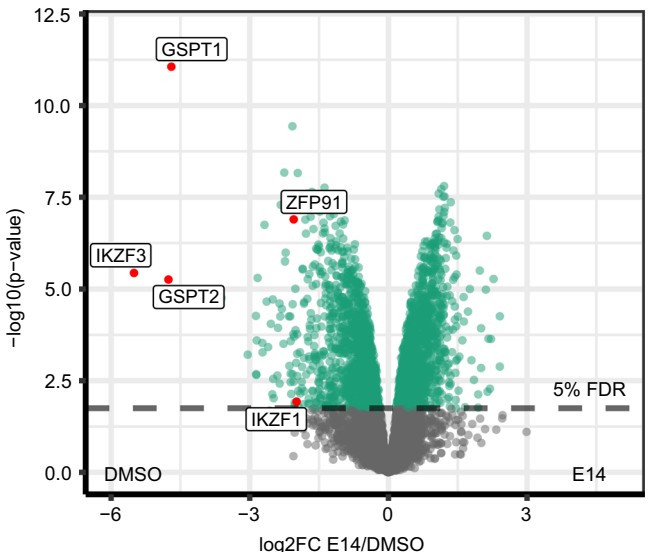

**Fig. 9 | Quantitative proteomics analysis of compound E14.** Quantitative proteomics in MM.1S cells after 18 h of treatment with compound E14 at 1.0 μM concentration. The control (DMSO) and E14-treated MM.1S were compared using the moderated 2-sample sided 2-sample *t*-test (*n* = 4). For each protein the log10 *p*-value (*y*-axis) is plotted against the log2FC E14/DMSO. *P*-values were adjusted using the Benjamini-Hochberg procedure. The significance cut off (5% FDR) is indicated by color. Source data are provided as a Source Data file.

cloned into pLKO5.hU6.sgRNA.dTom lentiviral vector (gifts from D. Heckl, Hannover Medical School). The sgRNA sequences are as follows: hCRBN #1: forward-CACC GTCCTGCTGATCTCCTTCGC, reverse-AAAC GCGAAGGAGATCAGCAGGAC; hCRBN #3: forward- CACC GGATTCA-CATAAGCTGCCAT, reverse- AAAC ATGGCAGCTTATGTGAATCC. The knockout was confirmed by western blot.

### Cell viability assay

Cells were seeded in 96-well plates with indicated concentrations of respective compounds and plates were incubated at 37 °C for 96 h. Readout was performed with CellTiterGlo® Luminescent Cell Viability Assay (Promega) according to the manufacturer's protocol and luminescence was measured with POLARStar Omega plate reader (BMG LabTech). All results were normalized to non-treated conditions and data represent mean ± SD of biological triplicates.

### Antibodies

Primary antibodies used for Western blotting from Cell Signaling (Danvers, USA) include IKZF3 (clone D1C1E, #15103, RRID: AB_2744524, 1:1000), IKZF1 (clone D6N9Y, #14859, RRID: AB_2744523, 1:1000), IRF4 (clone D43H10, #4299, RRID: AB_10547141, 1:1000), c-Myc (clone D84C12, #5605, RRID: AB_1903938, 1:1000), eRF3(#14980, RRID: AB_2798677, 1:1000), Cas9(clone 7A9-3A3, #14697, AB_2750916,1:1000); antibodies from Sigma-Aldrich (St. Louis, USA) include anti-alpha-Tubulin (#T5168, RRID: AB_477579, 1:7000), CRBN(# HPA045910, RRID: AB_10960409, 1:1000). Secondary antibodies used for Western blotting from Cell Signaling include anti-rabbit IgG HRP-linked antibody (#7074, 1:5000), anti-mouse IgG HRP-linked antibody (#7076, 1:5000).

### Immunoblot assay

Cell pellets were lysed in IP lysis buffer (Pierce) containing HALT protease and phosphatase inhibitor cocktail (Thermo Scientific). Protein concentrations were determined using the BCA Protein Assay Kit (Thermo Fisher Scientific). Protein lysates were run at constant voltage on Sodium dodecyl sulfate-polyacrylamide gel electrophoresis (SDS-PAGE). Proteins were transferred onto Immobilon-P transfer

membranes (Millipore) at a constant amperage. Membranes were blocked in 5% non-fat dry milk (Santa Cruz) for 1 h and incubated with Primary antibodies diluted in 5% BSA at 4 °C overnight and then incubated with secondary HRP-conjugated antibodies diluted in 5% milk in TBST. Detection of proteins was performed using either WesternBright ECL HRP substrate or WesternBright Sirius HRP substrate (Advansta). Chemiluminescence was detected with ChemiDoc™ XRS+ System (Bio-Rad). Quantification was performed using ImageJ (National Institutes of Health).

### Proteomics analysis assay

Cell pellets were lysed with 8 M urea lysis buffer for 15 min at 4 °C (8 M urea, 50 mM Tris pH 8, 150 mM NaCl, 1 mM 2-chloroacetamide), supplemented with protease inhibitors (2 μg/ml aprotinin, 10 μg/ml leupeptin, 1 mM phenylmethylsulfonyl fluoride) as described before (Mertins et al). Lysates were then clarified by centrifugation (20,000 g, 15 min, 4 °C). Disulfide bonds were reduced (5 mM dithiothreitol for 1 h) and alkylated (40 mM chloroacetamide for 45 min in the dark). Afterwards samples were diluted 1:4 with 50 mM Tris-HCl pH 8 and digested using sequencing grade LysC (Wako Chemicals) for 2 h in weight-to-weight ratio of 1:50. Finally sequencing grade trypsin (Promega) was added at a weight-to-weight ratio of 1:50, and digestion was carried out overnight. Samples were acidified with formic acid (FA) followed by centrifugation (20,000 g, 15 min). The supernatant was further processed using Sep-Pak C18 cc Cartridges (Waters) for desalting.

For the LC/MS analysis, 1 μg of desalted peptides was utilized for each sample. Peptide were separated on a Vanquish Neo System (Thermo Fisher) with a gradient lasting 106 min and a flow rate of 250 ul/min. The mobile phase B was gradually increased from 4% to 20% over the first 67 min, then to 30% over the next 20 min, followed by 60% for 10 min, 90% for 5 min, and finally 0% for 2 min.

MS data was acquired on an Exploris 480 (Thermo Fisher) using data-independent acquisition (DIA) mode. Full scans were obtained at a resolution of 120,000, scanning a range of 350–1650 m/z. The maximum injection time (IT) was set at 20 ms, with an automatic gain control (AGC) target value of 3e6. Subsequent to the full scan, narrow isolation windows were used, covering the range of 375-1430 m/z (Table SX), acquired at 30,000 resolutions. The fixed first mass was set at 200 m/z, with an AGC target value of 300e6 (3000%), and a maximum IT of 54 ms. The normalized collision energy was set in stepped mode at 26%, 29%, and 32%. Dynamic exclusion was employed for 30 s, and ions with charge states of 1, 6, or higher were excluded from fragmentation.

Raw data was searched using DIA-NN 1.8.1 software against the human UniProt reference proteome (Demichev et al.). Library-free mode was used, with the in silico FASTA digest parameter enabled. The peptide length range was set to 7–30, and the precursor charge range was set to 1–4. The m/z range for precursors was 340–1650, and for fragment ions, it was 200–1800. The 'match between runs' parameter was enabled.

LFQ protein intensities from the DIA-NN pg output table were log2 transformed, filtered for valid values (>70%) and contaminants.3. The resulting intensities were median normalized, and missing values were imputed from a normal distribution with a downshift (-1.8 SD from the mean and the distribution width is 0.3 SD). Comparative analysis of experimental groups was conducted using a two-sided moderated two-sample *t*-test. The resulting *p*-values were corrected using the Benjamini-Hochberg method. The data analysis was performed using R (4.3.1).

### Statistical analysis and reproducibility

The data are presented as mean ± standard deviation (SD). Statistical analyses were performed by Microsoft Excel (version 16.66), R (4.3.1), Image J (version v1.5j8), Graphpad Prism (version 9.2.0), MestReNova

(version 14.1.0-24037). Each value determined from the dose–response curve was calculated using GraphPad Prism (version 9.2.0). All experiments were repeated more than twice, and the number of replications is described in the figure legends.

## Data availability
The mass spectrometry proteomics data generated in this study have been deposited in the ProteomeXchange partner repository database under accession code PXD046752. The SFC-mass data, cell viability data and western blot data generated in this study are provided in the Supplementary Information/Source Data file. Source data are provided with this paper.

## Code availability
Code could be found in our published papers, doi.org/10.1039/D0GC00363H (Hadian, M.; Shaabani, S.; Patil, P.; Shishkina, V. S.; Böltz, H.; Dömling, A. Sustainability by design: automated nanoscale 2, 3, 4-trisubstituted quinazoline diversity. *Green chemistry* **2020**, *22* (8), 2459-2467.).

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

## Acknowledgements

Funded in part by the European Union under ERA Chair ACCELERATOR project no. 101087318 (to A.D.). Views and opinions expressed are however those of the author(s) only and do not necessarily reflect those of the European Union or the European Research Executive Agency. Neither the European Union nor the granting authority can be held responsible for them. J.K. was supported by the Deutsche Forschungsgemeinschaft (DFG, Emmy-Noether Program Kr3886/2-2 and SBF-1074). Z.W. and X.G. acknowledge the China Scholarship Council for supporting.

## Author contributions

A.D. and J.K. designed the project and acquired funding. Z.W. and S.S. performed the nano-scale synthesis and resynthesis. Z.W. performed the SFC mass analysis. X.G. performed the phenotypic screening and cellular biology experiments. Y.L.D.N. performed the degradation experiment. V.S. and P.M. performed the proteomics analysis. Z.W. and X.G. analyzed the data. All authors contributed to the writing of the manuscript.

## Competing interests

The authors declare no competing interests.
