## [Peer Review File · Nature Communications]

REVIEWER COMMENTS

Reviewer #1 (Remarks to the Author):

In this manuscript, Wang et al. describe an automated and nano-scale approach for synthesizing pomalidomide-based compounds. The authors consider that they can generate novel molecular glues (MGs) using such a synthetic approach and they claim to have a "direct-to-biology" platform because they can combine the nano-scale high-throughput synthesis with phenotypic screening. The authors describe one compound (E14) that they consider outperforms IMiDs that are approved and used in the clinic, because it kills cells that are IKZF1/3-independent. E14 and most of the "more potent" compounds, induce the degradation of GSPT1, a target essential in virtually all cell lines and which is degraded by several IMiDs already described (e.g., CC-885).

I appreciate the efforts of the authors and the nano-scale I.DOT-enabled synthesis could be valuable if the efficiency improves, but I believe that this manuscript is better suited for a more chemistry-focused journal. In addition, most of the claims are overstated, and the characterization of the small molecules is very poor (selectivity, ternary complex formation, etc. are lacking).

Major concerns

1. Chemistry: To generate a library of compounds with chemical diversity around pomalidomide, isocyanide-based multicomponent reactions were used. In particular, the Ugi reaction is one of the typical multicomponent reactions, hence I do not see much chemistry novelty.

2. Selectivity of the further studied compounds (e.g., E14) are not shown. Quantitative expression proteomics should be used. Otherwise, how are the authors sure that their molecules only destabilize IKZF1/3 and GSPT1? Drug-induced ternary complex formation is not assessed/proved either.

3. Conceptually, the authors claim that they are generating MGs (typically, monovalent), but indeed they are not decorating the pomalidomide entity but rather appending a "linker" to its primary amine and adding different building blocks in there (hence, far away from the pomalidomide core). This is not discussed. Does (e.g.) E14 really have a MG-like mode of action similar to pomalidomide (e.g., are the binary affinities and the cooperativity different/similar)?

4. Low efficiency: One 384-well plate (384 compounds) was attempted according to fig. 2. The reaction quality was analyzed using a previously published MS protocol and then analyze with an "in-house program" (this should be described in the methods or referenced if published). >40% of the reactions failed and for the rest, most of the wells had really low amounts of product.

5. 19 compounds were re-synthesized, which exact criteria were applied to select them? The explanation provided in lines 144-145 is very vague. In Fig. S10 the time and dose should be indicated. Cell viability could be compromised by indirect effects unrelated to the chemistry of the compound. The degradation potency (DC50, Dmax) should be explored.

6. Fig. 5: The only difference that these compounds have compared to pomalidomide is that they degrade GSPT1 as well (as many other compounds already described in the literature, e.g., CC-855).

7. It would be useful to include docking studies that help explain why E14, F9, etc. also degrade GSP1 in addition to IKZF1/3.

8. The introduction lacks many key references, and line 54 states that the mmol scale synthesis of MGs/PROTACs is "unsustainable". This is very subjective, and proved wrong by all the literature published so far on MG and PROTAC synthesis using mmol scales...

Minor points

1. Fig.4: EC50s would be more correct than IC50s. 2-dose viability "curves" are misleading (and non-fitted). (Fig. 4c, Fig. 6a). Bar graphs are more adequate.
2. Abbreviations are missed and used (or not) indiscriminately along the text. For example in the abstract: molecular glue or MG. Several
3. It would be ideal to review the text more carefully, there are many mistakes (e.g., line 28 "TBD" instead of TPD, line 36 "neosubstrates" where it should be "neosubstrate", etc.)

Reviewer #2 (Remarks to the Author):

This manuscript describes a proof of concept D2B approach to molecular glue discovery that combines an on-plate automated chemical synthesis platform with a target and ligase-focused phenotypic screening conducted in a MM.1S cell-based model. The 6 most active hits identified were selected and characterized resulting in the identification of E14 as the most potent compound.

The abstract and manuscript are clear and concise. The manuscript does not have any significant flaws. However, in my opinion some of the conclusions drawn from the results would need to be re-phrased in order to avoid misleading readers about their true relevance. Specifically, the following statements in the abstract could be revised:

- The phenotypic screen is presented in the abstract as "information-rich and high throughput". I imagine "information rich" refers to the potential to use the screen to access a large number of potential molecular mechanisms of action having an impact on substrate degradation. Maybe this could be made more clear. The phenotypic screen is not high throughput, at least in this study, even if it has the potential to be high throughput.

- The following statement in the abstract should also be revised: "...outperformed approved immunomodulatory imide drugs by several orders of magnitude". From the results shown E14 improves the potency of pomalidomide by 5-10 fold and Lenalidomide by 70 – 250 fold (much of which is the merit of using Pomalidomide as the starting point). The potency is certainly improved, but in the absence of additional selectivity, physicochemical, or DMPK data this looks like an overstatement.

The approach and results presented are original in that they show how existing methodologies (i.e. high throughput chemical synthesis and cell-based phenotypic screens) can be effectively combined for the discovery of new molecular glues in a way that can be more efficiently scaled and "industrialized" than other approaches.

This methodology appears suitable for the optimization of existing hits or chemical starting points in focused screens when new chemistry needs to be generated, as exemplified in this work with the pomalidomide pharmacophore. For more conventional screens, where exploring diversity is the priority, existing diverse chemical libraries can already provide access to relatively purified molecules in high numbers.

From a methodological perspective the work presented by Wang et al is of interest not only to scientists working on the discovery of novel molecular glues. It could also have potential applications for the optimization of phenotypic hits and also in conventional target-based drug discovery when using functional cellular assays during the hit-to-lead or optimization phases. In addition to the potential value of the methodological strategy described in this work, the molecules identified could be used as chemical starting points to generate more optimized compounds with improved potency, selectivity, or other desired drug-like properties.

One interesting result of this work is the discovery of inhibitory activity over GSPT1, an activity not present in Pomalidomide. This is very relevant as it illustrates the potential of using an existing glue to develop additional derivatives with novel activities, and the possibility offered by phenotypic screening to uncover novel unexpected activities even in such a focused screen, based on the use of a cell line depending on IKZF1/3 for proliferation.

Potential follow-up activities that could derive from this work could be the optimization of E14 and other representative compounds using SAR to identify the structural features that are associated with the increased potency and the degradation of an additional substrate like GSPT1. Optimized

compounds could be further profiled using transcriptomics and general selectivity panels to better benchmark the potential of these molecules vs Pomalidomide and other existing molecular glues. Referenced literature appears correct and includes representative and relevant papers, including those reviewing screening strategies previously used to discover new molecular glues (i.e. Domostegui et al. Chem Soc Rev. 2022 Jul 4;51(13):5498-5517).

Specific comments

Cell viability is shown as “%” with “1” as the activity of the DMSO controls. This is misleading. The suggestion is to change the legend to “Relative cell viability” or to change the scale to show 100% as the viability value of the control.

Figure S11 legend lacks information to understand what the panel and the color scale mean. In general, figure legends contain little and sometimes insufficient information that may be obvious to the authors but not to all readers. Reference to error bars and statistics is missing in figure legends.

The language used in lines 157 – 165 is confusing. An approach initially intended “To distinguish the effect from unspecified cell cytotoxicity” resulted in the identification of cytotoxicity in HEL and K562 cell lines, at least for some of the compounds, in particular E14. Verifying the lack of cytotoxic activity in cell lines not sensitive to degradation of IKZF1/3 as a surrogate of “phenotypic selectivity” appears to have been the initial objective which, by serendipity, led to the observation that there was indeed cytotoxicity due to degradation of GSPT1. For the sake of clarity, it would be better to indicate that this result was initially unexpected but still led to an interesting discovery. Phenotypic selectivity, or the low levels of non-specific cytotoxicity in cell types not sensitive to the CRBN-dependent activity of E14 at concentrations where the compound is active in control cells, is a relevant positive aspect that is shown in figure 6 where the viability of CRBN knockout MM.15 cells is only partially reduced at these concentrations. This should be acknowledged and highlighted by the authors as it demonstrates good “phenotypic selectivity” in a relevant cellular background.

The authors claim the cytotoxicity of E14 and other compounds in HEK and K562 is “explained” by the degradation of GSPT1 that they show in subsequent experiments. Unlike CRBN, where MM.1 knockouts are generated, this correlation is not conclusively demonstrated for GSPT1 as no CRBN knockouts are generated in HEL or K562 cells or any GSPT1-dependent cell line. Using CC-885 confirms that GSPT1 degradation impacts the viability of these cell lines but is not definitive proof that the inhibition of K562 viability induced by E14 is mediated by GSPT1 as this could also be the result of engaging other unknown off-targets in these cell lines. The recommendation is that this claim is changed to “could be explained” or moved to the discussion section as a likely hypothesis. Figure 1. It would be very helpful to have a diagram showing in a graphical easy to understand way that the screen went from 229 compounds to 15, 5/6, and ultimately to 1 (E14) as the top compound.

Figure 4. The concentration units should be clearly indicated in the legend of the dose responses in panel B. What do 1, 2, 3, 4, and 5 mean? 10, 100, 1000... nM?

Legend of figure 4. (A) “cell line” should read “cell lines”. (B) suggestion to change “anti-proliferative assay” to “anti-proliferative activity assay”. (C) suggestion to change “against HEL and K562 cells” to “in HEL and K562 cells”.

The legend of figure 6 should indicate that Luc1 and Luc2 refer to control cell lines without the CRBN knockout.

Conclusions

This manuscript shows the combination of a high throughput chemical synthesis method with a focused, yet valid, phenotypic approach that led to the identification of a potent Pomalidomide derivative. This molecule has the potential to be used as a starting point for further work, representing a relevant contribution to the field of molecular glues discovery.

In my opinion, in its current form, the manuscript could be suitable for publication, provided that the issues raised are addressed. Potential follow-up work could include SAR optimization aimed at identifying which structural features are responsible for the increased potency and GSPT1 degradation activity of E14, and conducting bioactivity profiling to assess their potential vs existing

molecular glues.

Reviewer #3 (Remarks to the Author):

This manuscript described an efficient platform for the discovery of molecular glues derived from well known cereblon E3 ligase ligands. The integration of high throughput automated nanoscale synthesis and phenotypic screening enabled the quick discovery of molecular glues with anti-proliferation activity.

Degraders such as molecular glues, PROTACS, and LYTACs are emerging novel modalities for drug discovery. There are ways to more rationally design bifunctional molecules such as PROTACs as the binding interactions of the two ligands with their targets are predictable, though the linker part still requires screening and optimization. However, it is challenging to design molecular glues. To date, most molecular glues are discovered serendipitously. A high throughput platform is definitely critical for the discovery of molecular glues.

The authors build on their expertise on high throughput nanoscale synthesis and multi-component reactions and applied them to the discovery of novel molecular glues in this work. Detailed mechanistic studies are also provided to support the proposed mechanism. Although degraders of IKZF1/3 and GSPT1 are not new, this platform will likely generate more interesting molecular glue degraders in the future. Overall, it is a timely and highly important work to the field of automatic synthesis and degrader development. This reviewer recommends the publication of this work after minor revisions suggested below.

The authors may consider the citation of some of the following papers on high throughput synthesis of PROTAC type of degraders under miniaturized conditions, which are related to the present study.

"Development of Selective FGFR1 Degraders using a Rapid Synthesis of Proteolysis Targeting Chimera (Rapid-TAC) Platform" Guo, L.; Liu, J.; Nie, X.; Wang, T.; Ma, Z.-X., Yin, D. Tang, W. *Bioorg. Med. Chem. Lett.* 2022, 75, 128982.

"A Platform for the Rapid Synthesis of Proteolysis Targeting Chimeras (Rapid-TAC) under Miniaturized Conditions" Guo, L.; Zhou, Y.; Nie, X.; Zhang, Z.; Zhang, Z.; Li, C.; Wang, T.; and Tang, W. *Eur. J. Med. Chem.* 2022, 236, 114317.

"Two-stage Strategy for Development of Proteolysis Targeting Chimeras and its Application for Estrogen Receptor Degraders" Roberts, B. L.; Ma, Z.-X.; Gao, A.; Leisten, E. D.; Yin, D.; Xu, W.; and Tang, W. *ACS Chem. Biol.* 2020, 15, 1487–1496.

ANSWERS TO REFEREES

Reviewer #1 (Remarks to the Author):

In this manuscript, Wang et al. describe an automated and nano-scale approach for synthesizing pomalidomide-based compounds. The authors consider that they can generate novel molecular glues (MGs) using such a synthetic approach and they claim to have a “direct-to-biology” platform because they can combine the nano-scale high-throughput synthesis with phenotypic screening. The authors describe one compound (E14) that they consider outperforms IMiDs that are approved and used in the clinic, because it kills cells that are IKZF1/3-independent. E14 and most of the “more potent” compounds, induce the degradation of GSPT1, a target essential in virtually all cell lines and which is degraded by several IMiDs already described (e.g., CC-885).

I appreciate the efforts of the authors and the nano-scale I.DOT-enabled synthesis could be valuable if the efficiency improves, but I believe that this manuscript is better suited for a more chemistry-focused journal. In addition, most of the claims are overstated, and the characterization of the small molecules is very poor (selectivity, ternary complex formation, etc. are lacking).

We thank the reviewers for the appreciation of our work and critical evaluation with very helpful advice for improvement.

Major concerns

1. Chemistry: To generate a library of compounds with chemical diversity around pomalidomide, isocyanide-based multicomponent reactions were used. In particular, the Ugi reaction is one of the typical multicomponent reactions, hence I do not see much chemistry novelty.

Answer: Although the Ugi reaction was reported in many places, the highlight of our project is using the I.DOT to build a molecule library in a rapid way and nano-scale. This platform is a good and innovative starting point for the hit compound screen. Meanwhile, the phenotypic screening was integrated into our platform, which could help to verify the active compounds quickly. This platform is useful not only for the MGs screening but also it has the potential to help the other drugs screening such as inhibitors. At the same time, other types of chemical reactions than Ugi reaction can be included and explored within this platform (see previous publications of us where we performed multiple heterocycle syntheses, Suzuki couplings etc). BTW most if not all medicinal chemistry projects reported in literature focus on technology and activity and not on novelty in chemistry. I am not aware about any recent medchem/technology publication exhibiting chemistry novelty, rather the great majority of medchem papers uses the beloved top pharma reactions (amidation, Suzuki etc) again and again.

2. Selectivity of the further studied compounds (e.g., E14) are not shown. Quantitative

expression proteomics should be used. Otherwise, how are the authors sure that their molecules only destabilize IKZF1/3 and GSPT1? Drug-induced ternary complex formation is not assessed/proved either.

Answer: The proteomics analysis was performed and included in the manuscript (page 13, figure 9). In brief, the proteomics study revealed potent degradation of IKZF3, GSPT1, and also its close homolog GSPT2 which has so far only rarely been described as IMiD neo-substrate. Ternary complex formation of IMiD, CRBN E3 ligase, and neo-substrates (including IKZF1/3, GSPT1) has been shown previously in several publications by co-immunoprecipitation and/ or crystal structures (PMID: 24292625; PMID: 24292625, PMID: 30385546; PMID: 30385546).

3. Conceptually, the authors claim that they are generating MGs (typically, monovalent), but indeed they are not decorating the pomalidomide entity but rather appending a "linker" to its primary amine and adding different building blocks in there (hence, far away from the pomalidomide core). This is not discussed. Does (e.g.) E14 really have a MG-like mode of action similar to pomalidomide (e.g., are the binary affinities and the cooperativity different/similar)?

Answer: Thank you for your interesting comments. Firstly, Molecular glues are the molecules that could bind to the E3 ligase (in our case: CRBN) to form the new hydrophobic surface, then recruit the neosubstrate to induce the ternary complex formation and protein degradation. However, the PROTACs need a specific binder for the target protein, which is not included as a part of our compounds. Secondly, our compounds were designed and synthesized based on the pomalidomide core, and in further biological validation assays, they also showed similar degradation profiles. Thirdly, CC885 as a known molecular glue, also contains a 'linker-like' moiety. From the co-crystal structures, this moiety fills a narrow hydrophobic pocket and increases the binding affinity with CRBN, inducing the GSPT1 degradation. CC885 has similar-sized modifications to our compounds. Moreover, the authors believe that often there is a fluid transition between glues and PROTACs, which could be also the case here. Experimental dissection of MOA according to glue vs PROTAC however is impossible and discussing this aspect is very speculative.

4. Low efficiency: One 384-well plate (384 compounds) was attempted according to fig. 2. The reaction quality was analyzed using a previously published MS protocol and then analyze with an "in-house program" (this should be described in the methods or referenced if published). >40% of the reactions failed and for the rest, most of the wells had really low amounts of product.

Answer: The reference to "in-house program" was added (Page 6, line 111-116, reference 12). We think a 60% success rate with unoptimized reaction conditions is still high when compared with literature reported reactions performed under realistic incorporation of (unprotected) functional groups. See for example the extensive efforts of industrial groups to find optimal catalyst/ligand systems for C-C coupling.

At the beginning of the project, the starting materials for the plate synthesis were not screened. We just checked our chemical storage and used what was present. On the other hand, to keep the product diverse, many starting materials that normally performed poorly in reaction were included. Finally, we got a 60% reaction success rate, which we think it is a good success rate. Meanwhile, considering that using IDOT technology, a compound library could be built rapidly and efficiently and many actives were found.

5. 19 compounds were re-synthesized, which exact criteria were applied to select them? The explanation provided in lines 144-145 is very vague. In Fig. S10 the time and dose should be indicated. Cell viability could be compromised by indirect effects unrelated to the chemistry of the compound. The degradation potency (DC_{50} , D_{max}) should be explored.

Answer: The 19 compounds were chosen on the best activity in the phenotypic screening and variation of the modification. A more detailed explanation has been included (Page 8, lines 147-150). The time and dose have been added in Fig S10. We found that the compounds have no effect on cell viability in cells where CRBN has been genetically inactivated by CRISPR/Cas9 demonstrating that the binding to CRBN E3 ligase is essential for their activity and therefore supporting the proposed mechanism. The degradation potency (DC_{50}) was explored, and the result was included in Fig 8. Degradation potency assay showed E14 could lead to the loss of GSPT1 with $DC_{50} = 0.88$ nM, IKZF1 with $DC_{50} = 11.54$ nM, and IKZF3 with $DC_{50} = 13.84$ nM (D_{max} for IKZF1/3 and GSPT1 > 99%).

6. Fig. 5: The only difference that these compounds have compared to pomalidomide is that they degrade GSPT1 as well (as several other compounds already described in the literature, e.g., CC-855).

Answer: Through the proteomics analyses that are now included in the revised manuscript we found that E14 in addition to GSPT1 also degrades GSPT2 which is not targeted by CC-885. However, the main focus of this work is the description of the I.DOT to build a nano-scale molecule library in a rapid way in combination with phenotypic screening that we believe will be highly useful to find new active MGs, in a direct-to-biology approach.

7. It would be useful to include docking studies that help explain why E14, F9, etc. also degrade GSP1 in addition to IKZF1/3.

Answer: Several previous studies found that IMiD analogs can induce binding of CRBN to GSPT1 resulting in ubiquitination and degradation. The most important step for the neosubstrate degradation is the ternary complex. Crystallography would be a useful tool to understand. However, docking in a ternary complex is less accurate. Moreover, it is well established that Crbn binding and tertiary complex formation involves very large protein complex quaternary structure changes (see e.g. G.Lander's work @ Scripps). Thus, a simple Crbn docking would be meaningless and not helpful to understand the compounds activity. Meanwhile, again, the main focus is the description of the I.DOT technology in combination with phenotypic screening.

8. The introduction lacks many key references, and line 54 states that the mmol scale synthesis of MGs/PROTACs is "unsustainable". This is very subjective, and proved wrong by all the literature published so far on MG and PROTAC synthesis using mmol scales...

Answer: The more key references have been added. The unclear description has been changed and revised (Page 2, line 57-58).

Minor points

1. Fig.4: EC50s would be more correct than IC50s. 2-dose viability "curves" are misleading (and non-fitted). (Fig. 4c, Fig. 6a). Bar graphs are more adequate.

Answer: Thank you for your correction. The EC50 is more correct than IC50. Fig 4 and the manuscript have been revised. Bar graphs have used in the Fig 4c, Fig 6a, and Fig S12.

2. Abbreviations are missed and used (or not) indiscriminately along the text. For example in the abstract: molecular glue or MG. Several

Answer: All abbreviations have been rechecked and revised.

3. It would be ideal to review the text more carefully, there are many mistakes (e.g., line 28 "TBD" instead of TPD, line 36 "neosubstrates" where it should be "neosubstrate", etc.)

Answer: The manuscript has been checked carefully and mistakes have been revised and changed.

Reviewer #2 (Remarks to the Author):

This manuscript describes a proof of concept D2B approach to molecular glue discovery that combines an on-plate automated chemical synthesis platform with a target and ligase-focused phenotypic screening conducted in a MM.1S cell-based model. The 6 most active hits identified were selected and characterized resulting in the identification of E14 as the most potent compound.

The abstract and manuscript are clear and concise. The manuscript does not have any significant flaws. However, in my opinion some of the conclusions drawn from the results would need to be re-phrased in order to avoid misleading readers about their true relevance. Specifically, the following statements in the abstract could be revised:

- The phenotypic screen is presented in the abstract as "information-rich and high throughput". I imagine "information rich" refers to the potential to use the screen to access a large number of potential molecular mechanisms of action having an impact on substrate degradation. Maybe this could be made more clear. The phenotypic screen is not high throughput, at least in this study, even if it has the potential to be high throughput.

Answer: Thank you for this comment, we have rephrased the description to make it more clear and agree that the current experiments are not high throughput but show that they in principle are high throughput capable.

- The following statement in the abstract should also be revised: "...outperformed approved immunomodulatory imide drugs by several orders of magnitude". From the results shown E14 improves the potency of pomalidomide by 5-10 fold and Lenalidomide by 70 – 250 fold (much of which is the merit of using Pomalidomide as the starting point). The potency is certainly improved, but in the absence of additional selectivity, physicochemical, or DMPK data this looks like an overstatement.

Answer: We have rephrased the description to make it more clear in the abstract.

The approach and results presented are original in that they show how existing methodologies (i.e. high throughput chemical synthesis and cell-based phenotypic screens) can be effectively combined for the discovery of new molecular glues in a way that can be more efficiently scaled and "industrialized" than other approaches.

This methodology appears suitable for the optimization of existing hits or chemical starting points in focused screens when new chemistry needs to be generated, as exemplified in this work with the pomalidomide pharmacophore. For more conventional screens, where exploring diversity is the priority, existing diverse chemical libraries can already provide access to relatively purified molecules in high numbers.

From a methodological perspective the work presented by Wang et al is of interest not only to scientists working on the discovery of novel molecular glues. It could also have potential applications for the optimization of phenotypic hits and also in conventional target-based drug discovery when using functional cellular assays during the hit-to-lead or optimization phases. In addition to the potential value of the methodological strategy described in this work, the molecules identified could be used as chemical starting points to generate more optimized compounds with improved potency, selectivity, or other desired drug-like properties.

One interesting result of this work is the discovery of inhibitory activity over GSPT1, an activity not present in Pomalidomide. This is very relevant as it illustrates the potential of using an existing glue to develop additional derivatives with novel activities, and the possibility offered by phenotypic screening to uncover novel unexpected activities even in such a focused screen, based on the use of a cell line depending on IKZF1/3 for proliferation.

Potential follow-up activities that could derive from this work could be the optimization of E14 and other representative compounds using SAR to identify the structural features that are associated with the increased potency and the degradation of an additional substrate like GSPT1. Optimized compounds could be further profiled using transcriptomics and general selectivity panels to better benchmark the potential of these molecules vs Pomalidomide and other existing molecular glues.

Referenced literature appears correct and includes representative and relevant papers, including those reviewing screening strategies previously used to discover new molecular

glues (i.e. Domostegui et al. Chem Soc Rev. 2022 Jul 4;51(13):5498-5517).

Specific comments

Cell viability is shown as “%” with “1” as the activity of the DMSO controls. This is misleading. The suggestion is to change the legend to “Relative cell viability” or to change the scale to show 100% as the viability value of the control.

Answer: The figures have been modified to make them clear. “Relative cell viability” was used to make the figures more understandable. (Fig 4C, Fig 6A, Fig 7, Fig S12)

Figure S11 legend lacks information to understand what the panel and the color scale mean. In general, figure legends contain little and sometimes insufficient information that may be obvious to the authors but not to all readers. Reference to error bars and statistics is missing in figure legends.

Answer: We have removed Figure S11, as we believe it is not contributing to the understanding of the article’s message.

The language used in lines 157 – 165 is confusing. An approach initially intended “To distinguish the effect from unspecified cell cytotoxicity” resulted in the identification of cytotoxicity in HEL and K562 cell lines, at least for some of the compounds, in particular E14. Verifying the lack of cytotoxic activity in cell lines not sensitive to degradation of IKZF1/3 as a surrogate of “phenotypic selectivity” appears to have been the initial objective which, by serendipity, led to the observation that there was indeed cytotoxicity due to degradation of GSPT1. For the sake of clarity, it would be better to indicate that this result was initially unexpected but still led to an interesting discovery.

Answer: The manuscript has been revised to make it clear. (Page 8, line 163-171)

Phenotypic selectivity, or the low levels of non-specific cytotoxicity in cell types not sensitive to the CRBN-dependent activity of E14 at concentrations where the compound is active in control cells, is a relevant positive aspect that is shown in figure 6 where the viability of CRBN knockout MM.15 cells is only partially reduced at these concentrations. This should be acknowledged and highlighted by the authors as it demonstrates good “phenotypic selectivity” in a relevant cellular background.

Answer: Thank you for this comment. We highlighted more clearly now that the experiments in knockout cells reveal the dependence on CRBN confirming the proposed mode of action of the compounds.

The authors claim the cytotoxicity of E14 and other compounds in HEK and K562 is “explained” by the degradation of GSPT1 that they show in subsequent experiments. Unlike CRBN, where MM.1 knockouts are generated, this correlation is not conclusively demonstrated for GSPT1 as no CRBN knockouts are generated in HEL or K562 cells or

any GSPT1-dependent cell line. Using CC-885 confirms that GSPT1 degradation impacts the viability of these cell lines but is not definitive proof that the inhibition of K562 viability induced by E14 is mediated by GSPT1 as this could also be the result of engaging other unknown off-targets in these cell lines. The recommendation is that this claim is changed to "could be explained" or moved to the discussion session as a likely hypothesis.

Answer: It is possible that the cytotoxicity of E14 in HEL and K562 could be the result of engaging other potential and unknown neo-substrates in these cell lines. Our proteomic study and immunoblots revealed the strong degradation potency on GSPT1 and given its essential role in cell survival it is the most likely target leading to cell toxicity. The description has been revised to make it more rigorous (Page 9, line 183-187).

Figure 1. It would be very helpful to have a diagram showing in a graphical easy to understand way that the screen went from 229 compounds to 15, 5/6, and ultimately to 1 (E14) as the top compound.

Answer: The corresponding number has been added in the Fig 1.

Figure 4. The concentration units should be clearly indicated in the legend of the dose responses in panel B. What do 1, 2, 3, 4, and 5 mean? 10, 100, 1000... nM?

Legend of figure 4. (A) "cell line" should read "cell lines". (B) suggestion to change "anti-proliferative assay" to "anti-proliferative activity assay". (C) suggestion to change "against HEL and K562 cells" to "in HEL and K562 cells".

The legend of figure 6 should indicate that Luc1 and Luc2 refer to control cell lines without the CRBN knockout.

Answer: The units for Figure 4A have been added and 1, 2, 3, 4, 5 are the logarithmic expression for 10, 100, 1000... nM. All the legend corrections have been done.

Conclusions

This manuscript shows the combination of a high throughput chemical synthesis method with a focused, yet valid, phenotypic approach that led to the identification of a potent Pomalidomide derivative. This molecule has the potential to be used as a starting point for further work, representing a relevant contribution to the field of molecular glues discovery. In my opinion, in its current form, the manuscript could be suitable for publication, provided that the issues raised are addressed. Potential follow-up work could include SAR optimization aimed at identifying which structural features are responsible for the increased potency and GSPT1 degradation activity of E14, and conducting bioactivity profiling to assess their potential vs existing molecular glues.

Reviewer #3 (Remarks to the Author):

This manuscript described an efficient platform for the discovery of molecular glues derived from well known cereblon E3 ligase ligands. The integration of high throughput automated nanoscale synthesis and phenotypic screening enabled the quick discovery of molecular glues with anti-proliferation activity.

Degraders such as molecular glues, PROTACS, and LYTACs are emerging novel modalities for drug discovery. There are ways to more rationally design bifunctional molecules such as PROTACs as the binding interactions of the two ligands with their targets are predictable, though the linker part still requires screening and optimization. However, it is challenging to design molecular glues. To date, most molecular glues are discovered serendipitously. A high throughput platform is definitely critical for the discovery of molecular glues.

The authors build on their expertise on high throughput nanoscale synthesis and multi-component reactions and applied them to the discovery of novel molecular glues in this work. Detailed mechanistic studies are also provided to support the proposed mechanism. Although degraders of IKZF1/3 and GSPT1 are not new, this platform will likely generate more interesting molecular glue degraders in the future. Overall, it is a timely and highly important work to the field of automatic synthesis and degrader development. This reviewer recommends the publication of this work after minor revisions suggested below.

The authors may consider the citation of some of the following papers on high throughput synthesis of PROTAC type of degraders under miniaturized conditions, which are related to the present study.

“Development of Selective FGFR1 Degraders using a Rapid Synthesis of Proteolysis Targeting Chimera (Rapid-TAC) Platform” Guo, L.; Liu, J.; Nie, X.; Wang, T.; Ma, Z.-X., Yin, D. Tang, W. *Bioorg. Med. Chem. Lett.* 2022, 75, 128982.

“A Platform for the Rapid Synthesis of Proteolysis Targeting Chimeras (Rapid-TAC) under Miniaturized Conditions” Guo, L.; Zhou, Y.; Nie, X.; Zhang, Z.; Zhang, Z.; Li, C.; Wang, T.; and Tang, W. *Eur. J. Med. Chem.* 2022, 236, 114317.

“Two-stage Strategy for Development of Proteolysis Targeting Chimeras and its Application for Estrogen Receptor Degraders” Roberts, B. L.; Ma, Z.-X.; Gao, A.; Leisten, E. D.; Yin, D.; Xu, W.; and Tang, W. *ACS Chem. Biol.* 2020, 15, 1487–1496

Answer: Thank you for the evaluation and appreciation of our work. As recommended these papers have been added and cited in our manuscript. (Page 13, line 240-242)

REVIEWERS' COMMENTS

Reviewer #1 (Remarks to the Author):

This Reviewer appreciates the addition of selectivity data (proteomics) and some of the responses provided by Wang et al.

Regarding the small molecules described in the paper, overall, I mostly see a small molecule class that many others have thoroughly explored. In addition, there is no novelty with regard to the target space. The authors highlight that the molecule E14 degrades (in addition to IKZF1 and GSPT1), the homolog GSPT2 "which has so far only rarely been described as IMiD neosubstrate". GSPT2 has been described as a neosubstrate plenty of times, it has to do more with the cell lines used for proteomics (cells expressing GSPT2). Some examples here: [10.1182/blood.2019000789](https://pubmed.ncbi.nlm.nih.gov/3211182/); [10.1038/nature14610](https://pubmed.ncbi.nlm.nih.gov/3211038/); [10.1038/nature14610](https://pubmed.ncbi.nlm.nih.gov/3211038/); [10.7554/eLife.38430](https://pubmed.ncbi.nlm.nih.gov/32107554/), etc.

Besides, the authors neither explored a potential difference in the binding mode by docking E14 in protein-protein available X-rays (which is not difficult despite what they state in the response), nor provided any kinetic data to contribute to the understanding of this degrader type.

Nevertheless, the sticky point to me is that, as the authors stressed many times, the main contribution of this work is the description of the I.DOT to build a nano-scale molecule library, which is somewhat interesting but of modest impact and novelty, is still driving my lack of support for this manuscript.

Reviewer #2 (Remarks to the Author):

The authors have properly addressed all the concerns raised by this reviewer and implemented the corrections suggested.

Specifically, several overstatements in the conclusions and interpretation of the results have been removed or revised.

REVIEWERS' COMMENTS

We are grateful to all reviewers for their constructive comments which helped us to improve our manuscript.

Reviewer #1 (Remarks to the Author):

This Reviewer appreciates the addition of selectivity data (proteomics) and some of the responses provided by Wang et al.

Regarding the small molecules described in the paper, overall, I mostly see a small molecule class that many others have thoroughly explored. In addition, there is no novelty with regard to the target space. The authors highlight that the molecule E14 degrades (in addition to IKZF1 and GSPT1), the homolog GSPT2 “which has so far only rarely been described as IMiD neosubstrate”. GSPT2 has been described as a neosubstrate plenty of times, it has to do more with the cell lines used for proteomics (cells expressing GSPT2). Some examples here: 10.1182/blood.2019000789; 10.1038/nature14610; 10.1038/nature14610; 10.7554/eLife.38430, etc.

Besides, the authors neither explored a potential difference in the binding mode by docking E14 in protein-protein available X-rays (which is not difficult despite what they state in the response), nor provided any kinetic data to contribute to the understanding of this degrader type.

Nevertheless, the sticky point to me is that, as the authors stressed many times, the main contribution of this work is the description of the I.DOT to build a nano-scale molecule library, which is somewhat interesting but of modest impact and novelty, is still driving my lack of support for this manuscript.

Answer: Thank you for your valuable input to improve the manuscript. In this manuscript, the nano-scale synthesis and cell-based phenotypic screening were combined to offer a platform for the novel MGs development. Most of MGs were discovered by chance. Our work offers a possible strategy to accelerate the discovery of novel MGs. Since our compound don't contain a special binder for the degraded substrates, it obviously means that our compound doesn't work in the PROTACs way. Docking of the compounds in our opinion is not meaningful, as also elaborated in the last revision and response to referee letter. All compounds with a Crbn warhead can be docked into the pdb structures. What should be the gained knowledge, apart from having a colorful figure in the publication?

Reviewer #2 (Remarks to the Author):

The authors have properly addressed all the concerns raised by this reviewer and implemented the corrections suggested.

Specifically, several overstatements in the conclusions and interpretation of the results have been removed or revised.

Answer: Thank you for your nice and helpful suggestions.